# Alternative RNA splicing in the endothelium mediated in part by Rbfox2 regulates the arterial response to low flow

Patrick A Murphy[1†]*, Vincent L Butty[2], Paul L Boutz[1], Shahinoor Begum[1,3], Amy L Kimble[4], Phillip A Sharp[1,2], Christopher B Burge[2], Richard O Hynes[1,2,3]*

[1]Koch Institute for Integrative Cancer Research, MIT, Cambridge, United States; [2]Department of Biology, MIT, Cambridge, United States; [3]Howard Hughes Medical Institute, United States; [4]Center for Vascular Biology, UCONN Health, Farmington, United States

**\*For correspondence:**
pamurphy@uchc.edu (PAM);
rohynes@mit.edu (ROH)

**Present address:** [†]Center for Vascular Biology, UCONN Health, Farmington, United States

**Competing interests:** The authors declare that no competing interests exist.

**Abstract** Low and disturbed blood flow drives the progression of arterial diseases including atherosclerosis and aneurysms. The endothelial response to flow and its interactions with recruited platelets and leukocytes determine disease progression. Here, we report widespread changes in alternative splicing of pre-mRNA in the flow-activated murine arterial endothelium in vivo. Alternative splicing was suppressed by depletion of platelets and macrophages recruited to the arterial endothelium under low and disturbed flow. Binding motifs for the Rbfox-family are enriched adjacent to many of the regulated exons. Endothelial deletion of *Rbfox2*, the only family member expressed in arterial endothelium, suppresses a subset of the changes in transcription and RNA splicing induced by low flow. Our data reveal an alternative splicing program activated by Rbfox2 in the endothelium on recruitment of platelets and macrophages and demonstrate its relevance in transcriptional responses during flow-driven vascular inflammation.
DOI: https://doi.org/10.7554/eLife.29494.001

## Introduction

Exposure of arterial endothelium to low and disturbed flow initiates a process of endothelial activation, resulting in increased expression of adhesion molecules and the rolling, adhesion and extravasation of leukocytes (*Gimbrone and García-Cardeña, 2013*; *Ley et al., 2007*). The level of shear stress to which endothelium is exposed is one of the best predictors of atherosclerosis and growth and rupture of aneurysms (*Boussel et al., 2008*; *Cheng et al., 2006*; *Cicha et al., 2011*; *Xiang et al., 2011*). Experiments with isolated endothelial cells exposed to altered flow in vitro revealed that increased laminar flow alone was sufficient to suppress endothelial apoptosis and expression of endothelial leukocyte recruitment molecules such as Icam1 (*Nagel et al., 1994*). Subsequent work revealed transcriptional programs induced by steady flow and suppressed by low and disturbed flow in vitro and in vivo (*Dai et al., 2004*). Many of these programs could be recapitulated by Erk5 activation and increased expression of the transcription factor Klf2 (*Dekker et al., 2006*; *Lin et al., 2005*; *Parmar et al., 2006*). Interfering with these programs increased markers of endothelial dysfunction and disease progression in atherosclerosis (*Dekker et al., 2006*; *Lin et al., 2005*; *Parmar et al., 2006*). Thus, exposure of arterial endothelium to low flow initiates a program of endothelial dysfunction characterized by transcriptional changes and alterations in the expression of leukocyte recruitment molecules.

Most work to date has focused on the primary endothelial response to low flow, rather than the chronic interactions with immune cells under low flow in vivo. However, chronic inflammation contributes to a wide variety of dangerous cardiovascular lesions, including the erosion and rupture of atherosclerotic plaques and aneurysms. Not surprisingly, inflammation is a clinical target in atherosclerosis (*Ridker et al., 2012*) and aortic aneurism (*Davis et al., 2014*). The adaptive immune response is also being targeted, with good efficacy in pre-clinical models of atherosclerosis (*Kimura et al., 2015*). Historical perspectives on inflammation have typically focused on the leukocytes recruited in large numbers from the blood to the tissues, however, the endothelial lining of arterial vessels plays a critical role as a regulator of multiple steps in the cascade, including effects on immune cell recruitment, trafficking into tissues, and local activity (*Carman and Martinelli, 2015*; *Galkina and Ley, 2009*; *Pober and Sessa, 2007*). The endothelium, as the interface between recruited immune cells and tissues, and as a local regulator of the inflammatory process, is an attractive target for the regulation of local inflammation, potentially avoiding complications of systemic immunosuppressive agents.

Recently, we discovered that, in a chronic in vivo model of low flow, changes in alternative splicing of the extracellular matrix protein fibronectin (FN) prevented hemorrhagic rupture of the intima (*Murphy and Hynes, 2014*). Through alternative splicing, a single pre-mRNA produced from a transcribed gene can be spliced in different ways to generate multiple, sometimes hundreds, of various isoforms (*Nilsen and Graveley, 2010*; *Wang and Burge, 2008*). These new isoforms often lend completely novel interactions to proteins, resulting in interaction profiles often as divergent from the 'canonical' isoform as from other unrelated proteins (*Yang et al., 2016*). Interestingly, the splicing changes we observed in *Fn*, resulting in the incorporation of alternative exons *EIIIA* and *EIIIB*, could be blocked by the depletion of macrophages, which are recruited to the arterial endothelium under low flow (*Murphy and Hynes, 2014*). Our results suggested that regulation of alternative splicing might play an important role in the endothelial interaction with recruited immune cells and regulation of chronic arterial inflammation.

Here, we test the possibility that alternative splicing is broadly regulated in the arterial endothelium by low and disturbed flow, and hypothesize that alternative splicing events co-regulated with alternatively spliced *Fn* exons *EIIIA* and *EIIIB* may be important in coordinating the response of the arterial endothelium to low flow.

## Results

### Low flow induces a program of alternative splicing in the arterial intima

To determine the extent of alternative splicing changes in the arterial intima under low-flow conditions, we prepared pools of intimal RNA from the carotid arteries of mice exposed to low flow for 48 hr (*Figure 1A*). Control pools of RNA were prepared from mice subjected to a sham operation, or from the contralateral arteries of ligated mice, which were exposed to high flow. We detected 4766 differentially expressed genes between the low-flow carotid and the contralateral high-flow carotid (Padj <0.05 by DEseq2, N = 3 per group), which overlapped and extended previous experiments in this system and in vitro (*Dai et al., 2004*; *Ni et al., 2010*) (*Figure 1B*). We observed excellent correlation in Log2 fold-change between biological replicates (Pearson correlation of 0.82 for all genes, and 0.98 for genes found to be significantly regulated by low flow). Low flow, rather than high flow in the contralateral artery, was responsible for the majority of changes in mRNA levels, relative to the normal-flow, sham-operated controls (*Figure 1—figure supplement 1*). The flow-responsive signature included reduction in expression of the canonical flow-responsive genes *Klf2* and *Klf4* and we also detected an increase in signal from innate immune cells (*CD45*, *F4/80*, *CD11b*, *Cxcr2*; *Figure 1—figure supplement 2*), suggesting that such cells are recruited as early as 48 hr after the change in flow. However, transcripts from recruited or other contaminating cells represented a small portion of the total signal. This was assessed by the amount of *eGFP* (endothelial) mRNA copies versus *tdTomato* (all other cells) in *Cdh5(PAC)Cre-ERT2; mT/mG* mice, in which *eGFP* or *tdTomato* are driven from the same constitutive actin promoter in a cell-specific manner (*Figure 1—figure supplement 3*). By this measure, contaminating transcripts represented <1.5% of RNA analyzed.

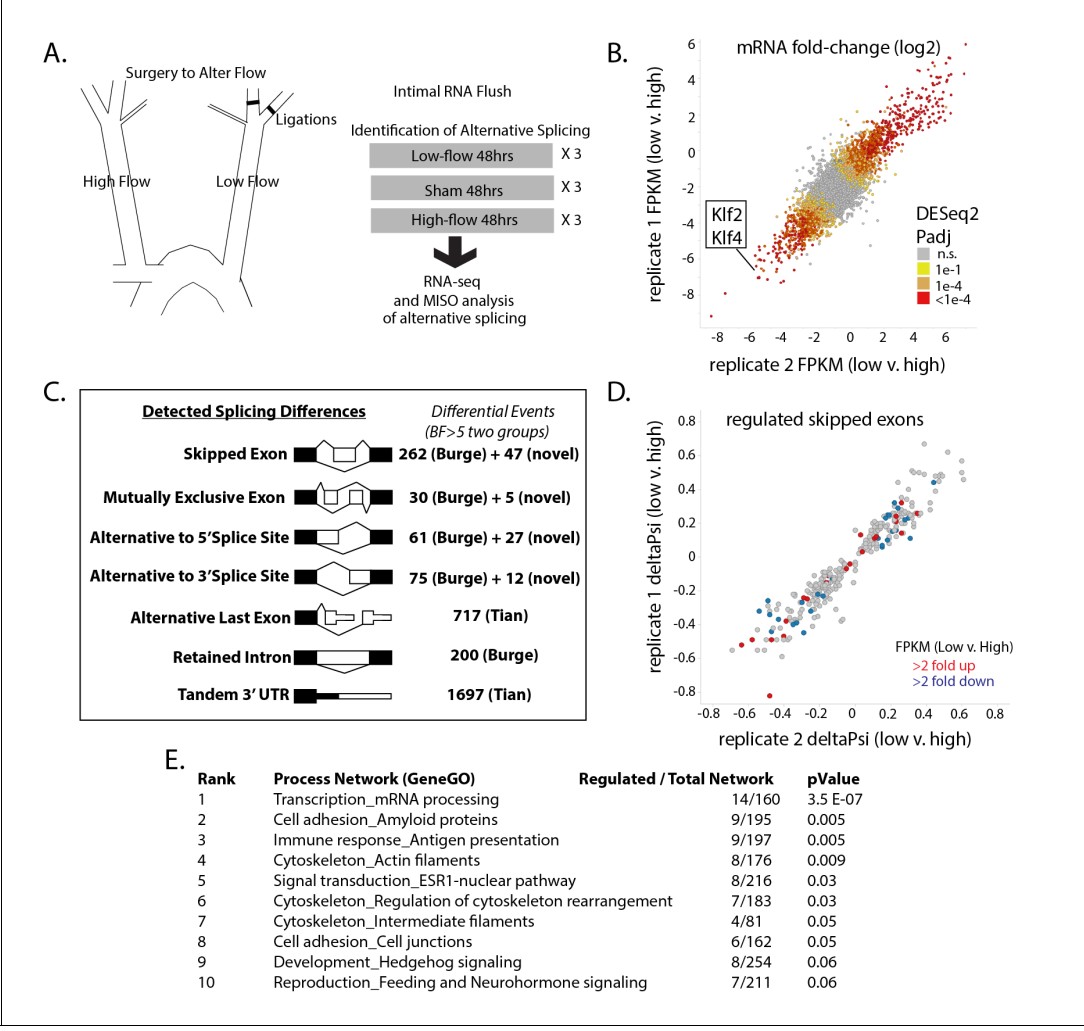

**Figure 1.** Exposure of the arterial endothelium to low and disturbed flow induces a program of alternative splicing. RNA from the arterial endothelium was isolated 48 hr after partial carotid ligation; from the low-flow side, the high-flow side, or sham-operated vessels. (**A**) Outline of splicing analysis. Three pools of mRNA from each condition were isolated by polyA and sequenced. (**B**) Plot showing the consistency of changes in gene expression in two independent biological comparisons of low-flow versus high-flow isolations. (**C**) Number of RNA splicing events of each category detected by MISO analysis as significantly different in two independent biological comparisons. Events were drawn from annotated databases (Burge, Tian) or from custom annotation of mapped splice junctions (novel). (**D**) Plot showing the changes in skipped-exon inclusion level (deltaPsi) between low-flow and high-flow isolations. The plot also indicates changes in transcription, highlighting exons in genes with a change in FPKM of more than 2-fold up or down. (**E**) Processes enriched in the genes with regulated skipped exons, relative to the entire set of genes expressed in the tissue with annotated skipped exons which were not significantly regulated. Padj = Adjusted P-value, BF = Bayes Factor; FPKM = Fragments Per Kilobase of transcript per Million mapped reads; CCDS = Consensus Coding DNA Sequence.

DOI: https://doi.org/10.7554/eLife.29494.002

The following source data and figure supplements are available for figure 1:

**Source data 1.** Contains DESeq2 output from the analysis of biological triplicates of low flow and biological triplicates of high flow, where each biological triplicate contained 4–5 carotid flushes.

DOI: https://doi.org/10.7554/eLife.29494.008

**Source data 2.** Contains MISO from Tophat alignments of 80 bp reads, as described in the methods.

DOI: https://doi.org/10.7554/eLife.29494.009

**Source data 3.** Contains bowtie alignments to eGFP or tdTomato for the indicated samples as described in the methods.

DOI: https://doi.org/10.7554/eLife.29494.010

**Figure supplement 1.** Gene expression changes between low-flow and high-flow intima are mainly due to low flow.

DOI: https://doi.org/10.7554/eLife.29494.003

**Figure supplement 2.** Hematopoietic cell recruitment within 48 hr of low and disturbed flow.

DOI: https://doi.org/10.7554/eLife.29494.004

*Figure 1 continued on next page*

*Figure 1 continued*

**Figure supplement 3.** Enrichment of endothelial RNA demonstrated by lineage markers in mT/mG mice.
DOI: https://doi.org/10.7554/eLife.29494.005
**Figure supplement 4.** Splicing changes are primarily induced by low flow rather than high flow.
DOI: https://doi.org/10.7554/eLife.29494.006
**Figure supplement 5.** Arterial cells cultured in vitro replicate splicing changes observed under low and disturbed flow in vivo.
DOI: https://doi.org/10.7554/eLife.29494.007

Having established endothelial enriched RNA pools from the various flow conditions, we then asked whether alternative splicing in intimal endothelial cells was regulated by flow. We assessed levels of alternative exon inclusion, expressed as 'percentage spliced in' (Psi), in each condition and their differences using the MISO Bayesian statistical framework (*Katz et al., 2010*). We examined annotated events as well as novel events (see Materials and methods). Setting a threshold of significance at Bayes factor five in two biologically independent comparisons of low flow to high flow, we identified 768 regulated splicing events between known exons (skipped or mutually exclusive exons, retained introns, or alternative 5′ or 3′ splice sites) and over a thousand regulated changes to the last exon or 3′UTR (*Figure 1C*). As with the transcript level changes, the vast majority of the splicing changes were induced by low flow, rather than high flow, in comparison with sham-operated vessels (*Figure 1—figure supplement 4*). Approximately, 90% of the splicing changes observed in vivo were recapitulated (at least 50% in the same direction) in purified endothelial cells in vitro, supporting the endothelial regulation of these splicing events (*Figure 1—figure supplement 5*). It is not clear whether the 10% of events not similarly regulated in vitro reflect differences in the in vitro system or transcript variants from the ~1.5% contaminating mRNA.

Because the mechanisms of regulation of skipped exons are best understood, and because we are interested in alternative splicing events similar to the skipped exons in *Fn*, we focused on this subclass of alternative splicing events (*Murphy and Hynes, 2014*). Changes in skipped exons were consistent between completely independent biological sets, and not obviously correlated with increased or decreased abundance in the transcriptional levels of these genes between conditions (*Figure 1D*, each point indicates a regulated splicing event, colored points indicate events in genes with altered transcript levels). The majority of the skipped exons were as strongly conserved evolutionarily as constitutive exons in placental mammals, and much more so than introns (data not shown). Genes with regulated skipped exons had annotated functions in several categories (*Figure 1E*). Thus, altered flow results in changes in RNA splicing of exons coincident with, but independent of, transcriptional changes.

## Platelets are required for regulation of a subset of flow-regulated skipped exons

We previously observed that monocyte recruitment was required for increased inclusion of *Fn-EIIIA* and *EIIIB* exons after 48 hr of low flow (*Murphy and Hynes, 2014*). Platelets and granulocytes are also recruited to arterial endothelium in regions of low and disturbed flow (*Chèvre et al., 2014*; *Huo et al., 2003*; *Massberg et al., 2002*). We depleted each of these recruited cell types using clodronate liposomes for macrophages (*Murphy and Hynes, 2014*), anti-GPIbα for platelets and anti-Gr1 for neutrophils (*Labelle et al., 2014*). Platelets were essential for the recruitment of both macrophages and neutrophils in our system (*Figure 2—figure supplement 1*). Platelet depletion resulted in suppression of *Fn-EIIIA* and –*EIIIB* relative to control treatments, and in widespread changes in a large number of other flow-regulated skipped exons (*Figure 2—figure supplement 2*). For the platelet-regulated skipped exons (about 1/3 of all flow-regulated skipped exons), platelet depletion causes low-flow arteries to cluster with high-flow arteries in their splicing profile rather than with IgG or untreated low-flow arteries (*Figure 2A*). Thus, platelets are necessary for the regulation of many of the low-flow regulated skipped exons we observed.

In vivo experiments were unable to discriminate between the effects of platelets and macrophages on the splicing response of endothelium, since platelet depletion suppressed macrophage recruitment. To address more directly the specific requirement for each blood cell type, we prepared a murine aortic endothelial cell line for in vitro co-culture experiments (*Figure 2—figure supplement 3*). We first examined *EIIIA* inclusion levels (*Figure 2B1*). Cultured cells in static conditions

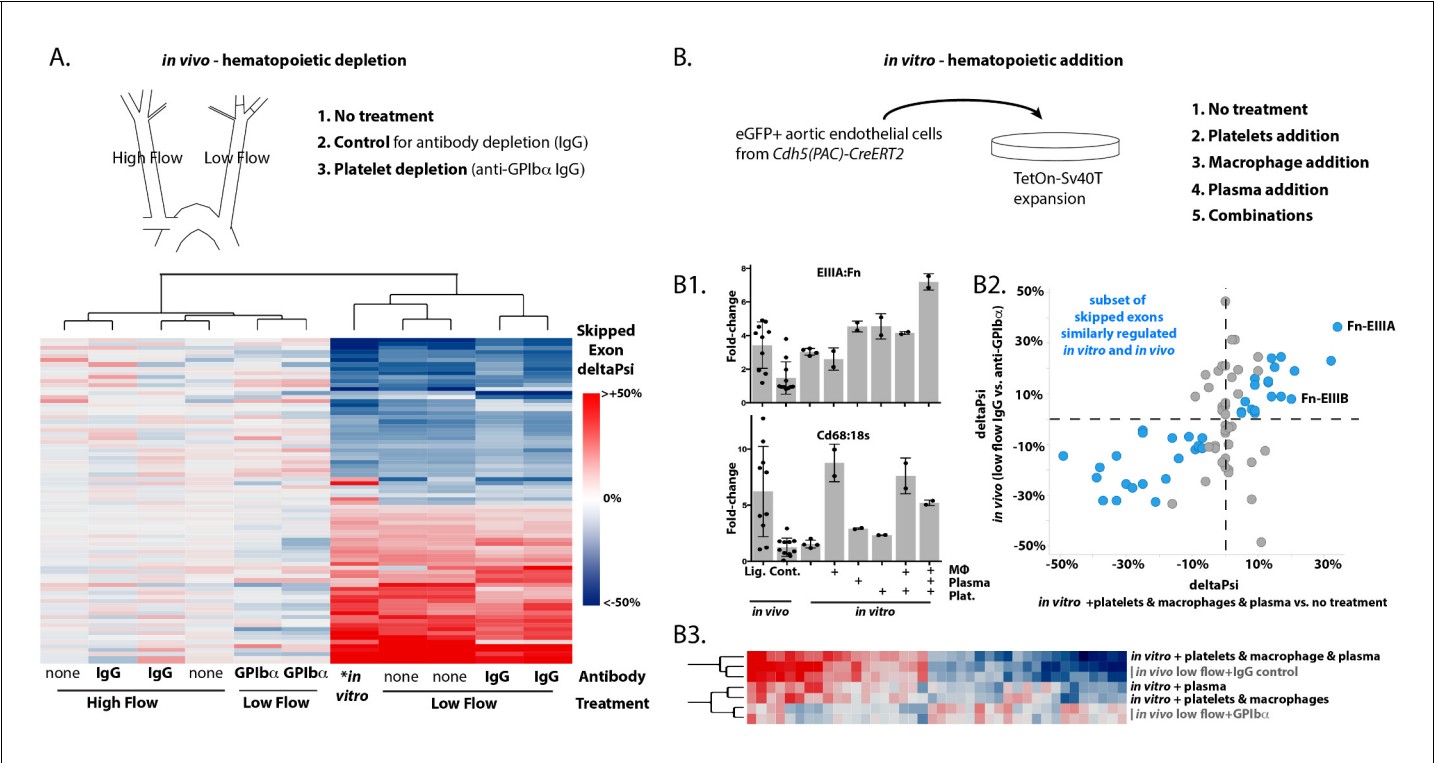

**Figure 2.** Platelets and macrophages regulate a subset of flow-regulated skipped-exon events. (A) Effects of platelet depletion on splicing patterns in vivo (see also 2–2). Clustered heat map showing the change in skipped-exon inclusion frequency relative to high-flow contralateral arteries in biological replicates of untreated arteries (from data set created in *Figure 1*) or platelet-depleted (anti-GPIbα) or IgG control-treated (IgG) arteries. The data shown are for the platelet-dependent subset of all flow-regulated skipped exons (SE; 80/292). IgG and anti-GPIbα (low flow) are relative to average IgG (high flow). No antibody treatment in vitro and in vivo low flow are relative to average in vivo high flow. (B) Effects of platelet, macrophage and plasma addition to endothelial cells in vitro. (B1) Bar graphs showing in vitro changes in EIIIA inclusion frequency and Cd68 macrophage marker expression with the different treatments of conditionally immortalized aortic endothelial cells, relative to in vivo low-flow and high-flow samples. (B2) Plot showing the in vitro regulation of platelet-regulated skipped-exon events (40/80), in isolated and conditionally immortalized aortic endothelial cells. (B3) Clustered heat map, showing the change in skipped exon inclusion frequency in in vivo biological replicates of low-flow arteries with or without platelet depletion (in vivo low-flow+ IgG control, +GPIba) or conditionally immortalized aortic endothelial cells with the addition of the indicated cells or 10% plasma.

DOI: https://doi.org/10.7554/eLife.29494.011

The following source data and figure supplements are available for figure 2:

**Source data 1.** Contains MISO from STAR alignments of 100 bp reads, as described in the methods.
DOI: https://doi.org/10.7554/eLife.29494.015

**Source data 2.** Contains ratio of *EIIIA* or *EIIIB* to total *FN* in the indicated samples.
DOI: https://doi.org/10.7554/eLife.29494.016

**Figure supplement 1.** Hematopoietic cell depletion in aortic intima 48 hr after induction of low and disturbed flow.
DOI: https://doi.org/10.7554/eLife.29494.012

**Figure supplement 2.** In vivo depletions of individual hematopoietic cell populations and their effects on Fn-EIIIA and –EIIIB inclusion.
DOI: https://doi.org/10.7554/eLife.29494.013

**Figure supplement 3.** Isolation and conditional immortalization of aortic endothelial cells.
DOI: https://doi.org/10.7554/eLife.29494.014

had higher baseline levels of *EIIIA* than high-flow arteries in vivo (compare bars 2 and 3). Co-culture with monocyte-derived macrophages, at similar stoichiometric ratios with the endothelial cells to those observed in vivo in low-flow arteries, did not further increase EIIIA inclusion. However, both platelets and plasma independently induced increased EIIIA inclusion, and exerted combinatorial effects, resulting in inclusion levels approximately 6-fold higher than in vivo levels when all three were added [*EIIIA* (from 15% to 80%) and *EIIIB* (from 11% to 60%)]. A broader examination of splicing changes, by RNA-seq, revealed that half (40 of 80) of the splicing events regulated by platelets

in vivo were similarly regulated by a combination of platelets, macrophages and plasma in vitro (*Figure 2B2*). For this set of events, the addition of plasma, macrophages and platelets to cultured endothelial cells induced a low-flow-like change in splicing but the individual components - platelets, macrophages or plasma proteins - were unable to entirely recapitulate the potent effects of their combination (*Figure 2B3*). Thus, platelets alone were insufficient to induce low-flow regulated skipped exons, despite static in vitro conditions, and required both macrophages and plasma for maximal effect.

Together our results suggest that recruitment of both platelets and macrophages to the arterial endothelium is required for the flow-responsive splicing pattern in a large set of genes, including the *Fn-EIIIA* and *EIIIB* exons.

## Binding motif for the splicing factor Rbfox2 is enriched in regulated vascular splicing events

We were particularly interested in the set of skipped exons induced in the endothelium in a platelet and macrophage-dependent manner, along with *Fn-EIIIA* and *–EIIIB*, and asked whether there might be a common upstream regulatory factor. Although analysis of transcriptional levels did not reveal obvious candidate splicing factors (*Figure 3—figure supplement 1*), the identity of relevant splicing regulators can in some cases be deduced by looking for enrichment of their RNA-binding motifs adjacent to regulated exons, where they typically bind to exert their effects (*Shapiro et al., 2011*).

To take an unbiased approach to the identification of regulatory splicing factors by flanking motifs, we developed an analysis pipeline using the ranking algorithm in GSEA and taking advantage of the recent development of a comprehensive database of RNA-protein binding preferences (CisBP-RNA) to provide the RNA-binding protein 'finger-prints' to be ranked (*Ray et al., 2013*). We confirmed that this pipeline was able to correctly identify known regulatory splicing factors from raw RNA-seq data from single-cell in vitro systems or complex in vivo tissues (*Figure 3—figure supplement 2*).

Having confirmed our pipeline, we then applied it to assess the regulation of skipped exons in the increasingly narrowly defined group regulated in vivo and in vitro by recruited platelets and macrophages (*Figure 3A and B*). Enrichment was judged relative to background sets created from known skipped exons expressed but not regulated in endothelial cells (see Materials and methods for details). The Rbfox family, of which Rbfox2 is known to regulate *EIIIB* (*Underwood et al., 2005*), stood out among this group, being enriched especially on the downstream side of exons with decreased inclusion and, to a lesser extent, on both the upstream and downstream side of exons with increased inclusion (*Figure 3C–E*).

Rbfox2 motifs in the flanking regions of the regulated events were conserved, consistent with the conservation of the exons themselves (*Figure 3—figure supplement 3*). The Rbfox family contains three members, Rbfox1-3, with conserved RNA-binding motifs. To determine which of these might be most important in regulation of the *EIIIA* and *EIIIB* cohort of events, we examined their expression in vivo, and in in vitro purified endothelial cells. We found that only *Rbfox2* was significantly expressed in arterial endothelial cells (*Figure 3F*).

Thus, bioinformatics analysis suggests *Rbfox2* as a particularly important regulator of the skipped exon changes induced by platelet and macrophage recruitment to the arterial endothelium under low flow conditions.

## Endothelial-specific deletion of *Rbfox2* partially reverts flow-responsive skipping of vascular skipped exons induced by low flow in vivo

To test whether *Rbfox2* is involved in the splicing response of the arterial endothelium to low flow, we genetically removed *Rbfox2* in mice using an endothelial-specific and inducible Cre (*Cdh5(PAC)-CreERT2*) in combination with floxed *Rbfox2* alleles (*Gehman et al., 2012*; *Sörensen et al., 2009*). This strategy resulted in efficient deletion of *Rbfox2* without an increase in expression of the other Rbfox family members, 1 and 3 (*Figure 3F*; EC-KO).

To allow us to assess both the low-flow-responsive splicing patterns and the chronic transcriptional response of the flow-activated intima, we examined the effect of *Rbfox2* deletion 7 days after the reduction in flow. Consistent with our previous observations on *Fn-EIIIA* and *–EIIIB*, we found that most splicing changes observed at 48 hr were also observed at 7 days (*Murphy and Hynes,*

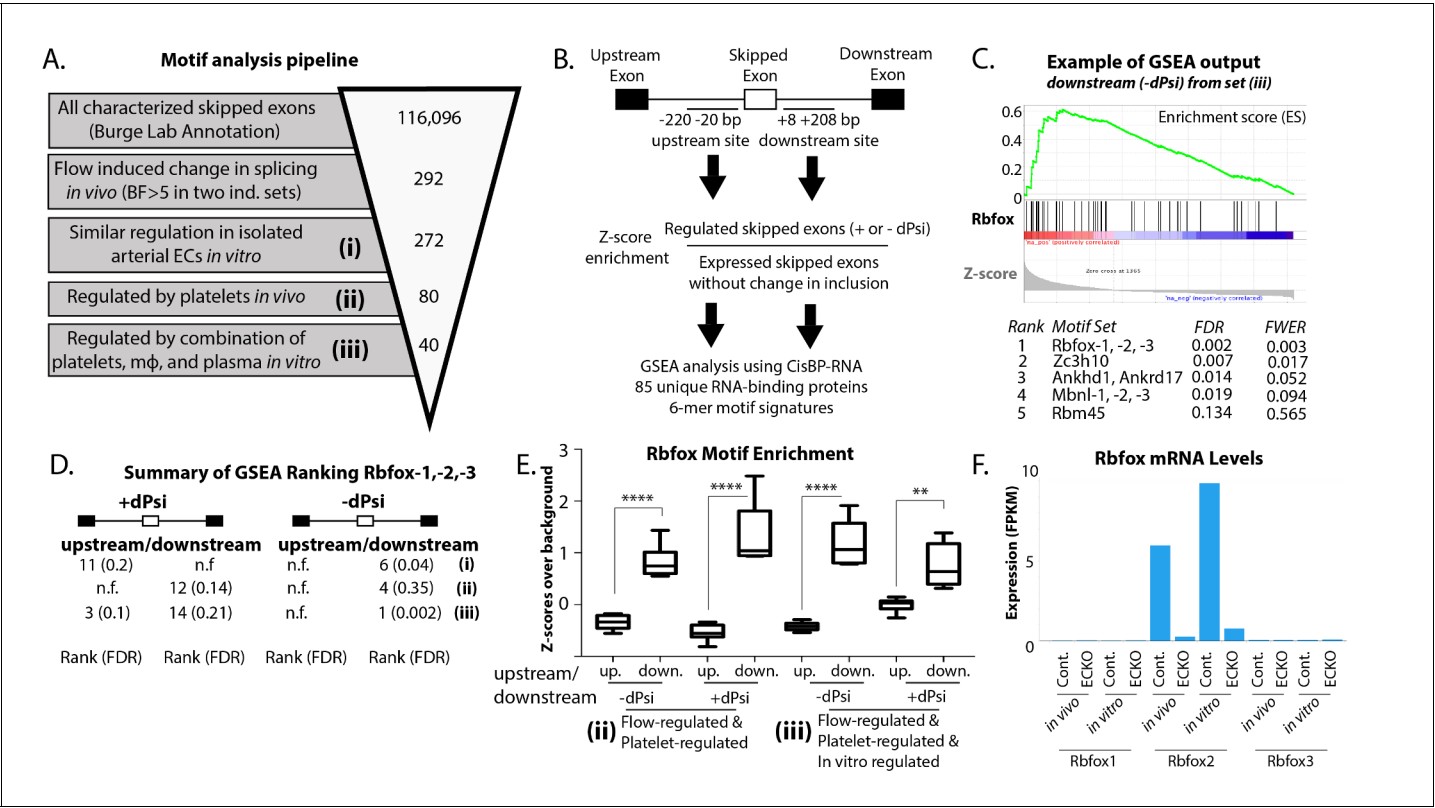

**Figure 3.** Enrichment of an Rbfox motif in the platelet/macrophage-regulated subset identified in vivo and in vitro. (**A**) Motif analysis pipeline, showing the number of potentially regulated skipped-exon events, and those passing each filter. (**B**) Events were separated into those with increased or decreased inclusion, and then the sequences of the upstream and downstream 200 bp flanking regions were isolated, resulting in four regions of analysis. 6-mer motifs in each region were assessed for enrichment relative to exons expressed in six matched background sets of skipped exons not regulated by flow. GSEA was used to identify splicing factor 'fingerprints' among these enriched motifs, using the motif-binding preference defined by the CisBP-RNA database (Ray et al.). (**C**) Example of enrichment of Rbfox binding motif in GSEA analysis of 85-unique RNA-binding protein signatures against flanking intron sequences of flow-regulated exons. Plot shows enrichment of Rbfox family motifs among the motifs most enriched above background in the downstream flanking region of exons regulated in vivo and in vitro (set iii). (**D**) Enrichment of the Rbfox2 motif in each of the sets of biologically defined regulated exons (i-iii) within the upstream and downstream flanking regions of exons with either increased or decreased inclusion. (**E**) Plot showing the average z-score enrichment of the top 5 Rbfox in vitro defined motifs in the flanking regions of the in vivo regulated exon set (80 SE) and the in vitro regulated subset (40 SE), relative to the six matched background sets (N = 6 per bar). (**F**) Transcript levels of Rbfox family of proteins in the carotid artery in vivo and in isolated aortic endothelial cells in vitro, with or without the deletion of *Rbfox2* by *Cdh5(PAC)-CreER* (EC-KO). FPKM = Fragments Per Kilobase of transcript per Million mapped reads. FDR; false-discovery rate; n.f.; not found. p<0.0001 (****), p<0.01 (**).

DOI: https://doi.org/10.7554/eLife.29494.017

The following source data and figure supplements are available for figure 3:

**Source data 1.** Contains the pared down list of flow-regulated events and their annotations (e.g. platelet regulated).
DOI: https://doi.org/10.7554/eLife.29494.021

**Source data 2.** Contains the list of intervals used for motif enrichment around the indicated regulated exons for the indicated sets (e.g. increased inclusion in flow regulated set).
DOI: https://doi.org/10.7554/eLife.29494.022

**Source data 3.** Contains a list of expressions across samples for all known RNA binding proteins.
DOI: https://doi.org/10.7554/eLife.29494.023

**Source data 4.** Contains the list of all motifs, their enrichment adjacent to regulated exons, their conservation score, and the z-score enrichment of the motif from the CisBP-RNA database.
DOI: https://doi.org/10.7554/eLife.29494.024

**Figure supplement 1.** No obvious candidate splice factors from differential expression under altered flow conditions.
DOI: https://doi.org/10.7554/eLife.29494.018

**Figure supplement 2.** Positive control sets to confirm the motif-based identification of regulated events in vitro and in vivo by their CisBP-RNA fingerprint in GSEA.
DOI: https://doi.org/10.7554/eLife.29494.019

*Figure 3 continued*

**Figure supplement 3.** Conservation of Rbfox2 motifs enriched near flow-regulated skipped exons.
DOI: https://doi.org/10.7554/eLife.29494.020

*2014*). We found that *Rbfox2* was important in the flow-responsive splicing change of many of the previously defined skipped exons, including *Fn-EIIIB* and *–EIIIA* to a lesser extent (*Figure 4A*, and

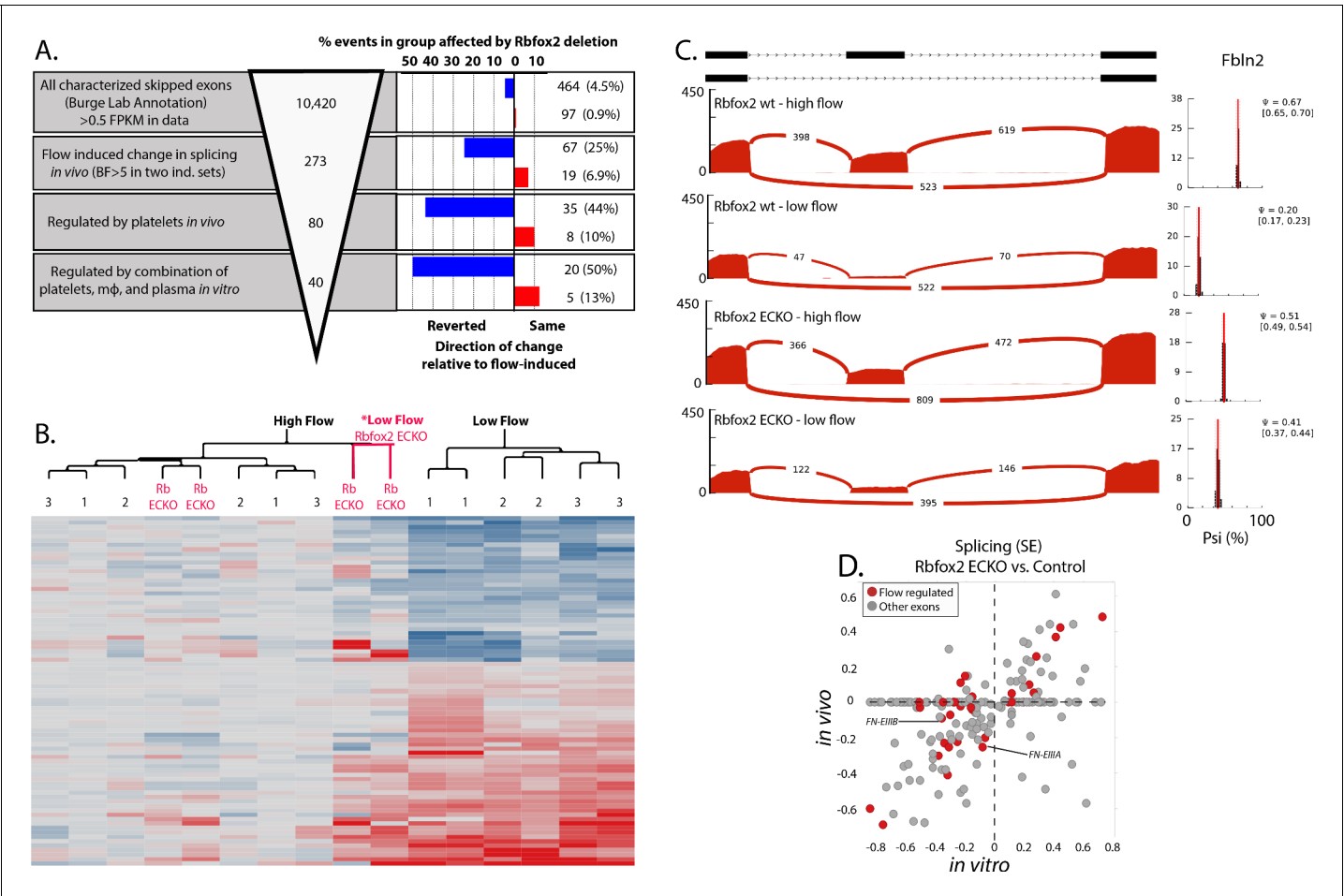

**Figure 4.** Endothelial Rbfox2 deletion affects many of the flow-regulated splicing events. (**A**) Showing the percentage of each group of skipped exons regulated by endothelial deletion of Rbfox2 (BF >5) in any comparison and regulated similarly in both comparisons of Rbfox2 wt low-flow vs. Rbfox2 EC-KO low-flow – either consistent with the flow-induced change in splicing (red, 'same') or against the direction of the flow-induced change in splicing (blue, 'reverted'). (**B**) Heat map showing the clustering of flow-regulated splicing events reverted by Rbfox2 deletion (67 of the 273 events consistently regulated between 48 hr and 7 days). 1 = C57 wild-type mice, 48 hr data set; 2 = C57 wild-type mice, IgG control 48 hr data set; 3 = Rbfox2 wt control (i.e. no Cre), 7 days data set; Rb EC-KO = Rbfox2 EC-KO 7 days data set. (**C**) Effect of endothelial *Rbfox2* deletion on flow-mediated regulation of inclusion of *Fbln2* alternative exons. (**D**) Plot shows the change in splicing of skipped exons (SE) following *Rbfox2* deletion from carotid artery intima in vivo (under low flow) or in isolated primary aortic endothelial cells in vitro. SE from the *Rbfox2* regulated set of events, in genes expressed >FPKM 1 in in vitro and in vivo sets. SE also regulated by a change in flow are highlighted.
DOI: https://doi.org/10.7554/eLife.29494.025

The following source data and figure supplements are available for figure 4:

**Source data 1.** Contains paired list of flow-regulated skipped exon events.
DOI: https://doi.org/10.7554/eLife.29494.028

**Figure supplement 1.** Rbfox2 deletion does not cause a reduction in markers of recruited platelets or macrophages.
DOI: https://doi.org/10.7554/eLife.29494.026

**Figure supplement 2.** Rbfox2 localization under serum stimulation in mouse aortic endothelial cells.
DOI: https://doi.org/10.7554/eLife.29494.027

*Figure 4—source data 1*). Altogether, *Rbfox2* deletion suppressed flow-induced splicing changes of ~25% the skipped exons (*Figure 4A*, blue bars, right hand panel and 4B). This set of genes encodes extracellular matrix proteins (e.g. *Fn1*, *Fbln2*, *Mfge8*) and immune-regulatory proteins (e.g. *Ikbkg*, *Gpr116*, *Ceacam1*). Analysis of inclusion of specific exons is shown in *Figure 4C* demonstrating the loss of flow-responsive exons of *Fbln2* on endothelial deletion of *Rbfox2*.

Since suppression of flow-responsive splicing was also impaired by depletion of platelets and macrophages, we asked whether recruitment of these cells was affected. However, we observed similar levels of macrophage markers (*Cd68, F4/80*) in the low-flow intimal flushes in the presence and absence of endothelial *Rbfox2* (*Figure 4—figure supplement 1*), suggesting that recruitment of macrophages, and by extension the platelets that we had previously found to be required for macrophage recruitment, were not detectably affected by deletion of *Rbfox2*.

Consistent with a direct effect of *Rbfox2* depletion on endothelial cells, we found a good correlation between the changes in splicing observed in vivo upon *Rbfox2* deletion and those observed in vitro with *Rbfox2* deletion in primary isolated aortic endothelial cells (*Figure 4D*). Notably, changes in splicing in vitro, in response to serum, did not correspond with an obvious alteration in the localization or abundance of Rbfox2 in the nucleus at the protein level (*Figure 4—figure supplement 2*).

Thus, *Rbfox2* deletion in the endothelium prior to the induction of low flow suppresses a large portion of flow-responsive alternative splicing in endothelial cells, downstream of platelet and macrophage recruitment.

## Endothelial deletion of *Rbfox2* alters the intimal response to low flow

We predicted that changes in the splicing response of the arterial intima caused by *Rbfox2* deletion would affect the response to low flow. To test this, we examined gene expression at the whole transcript level 7 days after the induction of low flow in *Rbfox2* EC-KO mice and littermate controls, and in the isolated endothelial cells. We found that endothelial deletion of *Rbfox2* significantly affected

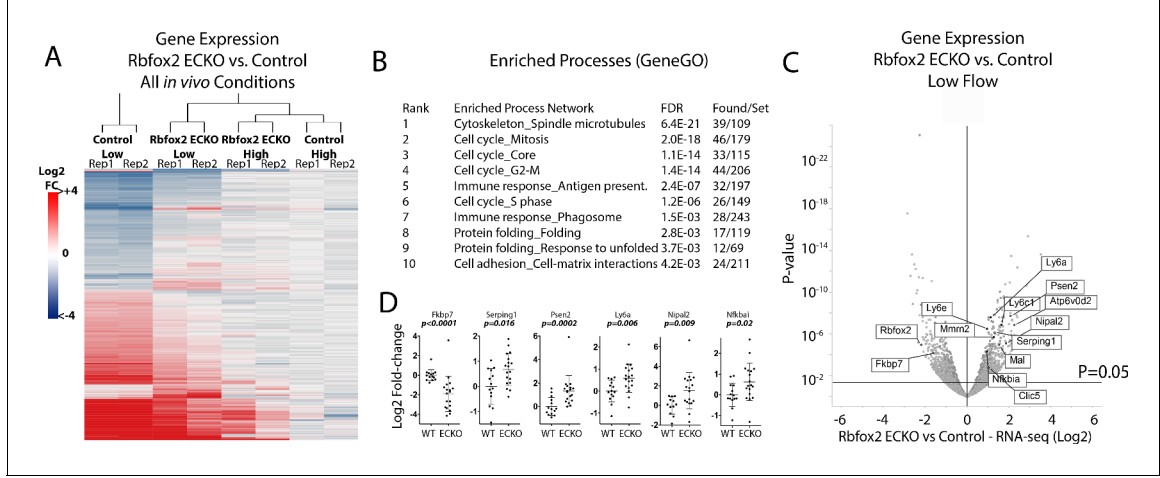

**Figure 5.** Endothelial Rbfox2 deletion suppresses low-flow transcript response in the arterial intima. (A) Clustered heat map of the genes with adjusted p-values<0.05 shows the change in expression relative to contralateral controls at 7 days after the change in flow in the indicated genotypes (N = 641). (B) Enriched terms among the genes regulated by Rbfox2 deletion. (C) Volcano plot showing DESeq2 calculated p-values and log2 fold changes in genes expressed in the arterial intima at 7 days of arteries exposed to low and disturbed flow, with or without deletion of *Rbfox2*. Genes selected for qPCR in single arteries are shown. (D) Results of individual carotid artery qPCR for the genes indicated. Log2 fold-changes are relative to control, p values are from Mann-Whitney test (N = 14 control and N = 18 Rbfox2 EC-KO).

DOI: https://doi.org/10.7554/eLife.29494.029

The following source data and figure supplement are available for figure 5:

**Source data 1.** Contains qPCR deltaCT between gene and housekeeping gene in the indicated samples, and the normalization to controls used to determine fold-change.
DOI: https://doi.org/10.7554/eLife.29494.031

**Figure supplement 1.** Rbfox2-dependent transcriptional response found in vivo is specific to the low-flow arterial intima.
DOI: https://doi.org/10.7554/eLife.29494.030

641 genes in the carotid intima under low flow (DESeq2 p-adj <0.05) (*Figure 5A*). Of these 641 genes, 507 are part of the 4766 gene low-flow signature previously described (*Figure 1*). Clustering of expression changes in the *Rbfox2*-regulated subset of genes, relative to the contralateral artery of control mice, shows that deletion of *Rbfox2* causes the low-flow arterial intima to resemble more closely the high-flow arterial intima (*Figure 5A*). Processes of cell proliferation and of antigen presentation were enriched among the differentially regulated genes (*Figure 5B*). RNA-seq was performed on pooled arteries, and quantitative PCR analysis of selected regulated genes in individual carotid arteries confirmed expression changes in individual animals (*Figure 5C and D*). These results indicate that Rbfox2 is essential for a portion of the endothelial response to low flow, regulating ~10% of the transcriptional response to low flow.

In contrast with our observation of correlations between splicing patterns in vivo and in vitro, most transcriptional changes induced in the low-flow arterial intima were specific to the in vivo environment and were not observed in a comparison of isolated aortic endothelial cells or in the contralateral artery (*Figure 5—figure supplement 1*). In other systems, cytoplasmic Rbfox has been shown to have an important role in the regulation of transcript stability through the 3'UTR (*Damianov et al., 2016*; *Lee et al., 2016*). However, transcript levels regulated by *Rbfox2* were not any more associated with characterized 3' UTR binding by iClip-Seq (*Lee et al., 2016*) than those not regulated by *Rbfox2* (iClip-Seq binding sites in 54/638 of the *Rbfox2*-regulated transcripts versus 1166/17130 of the transcripts not regulated by *Rbfox2*). Together, these data suggest that the consequences of endothelial Rbfox2 deletion on transcriptional changes involved cross-talk among cell types (e.g. effects of altered endothelial splicing on recruited immune cells) or other microenvironmental effects in the flow-activated intima in vivo not replicated in vitro.

Thus, *Rbfox2* deletion affects skipped exon splicing patterns in endothelial cells, and results in the reversion of a large set of low-flow-induced gene expression changes in the arterial intima.

## Discussion

Here, we report that recruitment of circulating hematopoietic cells to the arterial endothelium under low flow initiates an alternative splicing response regulated in part by *Rbfox2*. Profiling global transcriptional changes induced in the arterial endothelium by experimentally induced low flow, we show that hundreds of transcripts are regulated by alternative splicing. Focusing on skipped exons in particular, we find that one third of those affected by altered flow are dependent on recruited hematopoietic cells in vivo, particularly platelets. Rbfox-binding motifs are significantly enriched adjacent to these exons. Mechanistically, deletion of endothelial *Rbfox2* reverts many of the flow-induced changes in splicing and suppresses ~10% of the flow-induced changes in transcription. Processes related to genes with changes in either splicing or transcription included cytoskeleton, cell adhesion, proliferation and antigen presentation suggesting that *Rbfox2* may regulate these biological processes in the aortic intima downstream of hematopoietic cell recruitment.

### Broad regulation of alternative splicing by flow in the arterial endothelium

Low flow exerts potent effects on the vasculature through the endothelium, which are achieved by effects on endothelial transcription, epigenetic regulation, and post-translational protein modifications and reorganization (*Gimbrone and GarciaGarcía-CardenaCardeña, 2016*). Our work reveals widespread changes in alternative splicing affecting a number of genes known to be critical in the pathways already implicated in the response to altered flow, inflammation, remodeling and flow-driven vascular disease. Examples include *Pecam1*, a component of the shear-sensing mechanism in the endothelium (*Tzima et al., 2005*), *Yap1*, a regulator of mechanical activation (*Dupont et al., 2011*) and *Ikbkg (Nemo)*, a regulator of NFkappa-B signaling (*Li et al., 1999*). In previous work, differential splicing of *Vcam1* was detected in IL-1-stimulated HUVECs (*Cybulsky et al., 1991*) and of *Fn* in TGFβ-induced liver sinusoidal endothelium (*Chang et al., 2004*). Exon arrays have also shown differential exon usage in HUVECs in response to hypoxia (*Hang et al., 2009*; *Weigand et al., 2012*). However, this hypoxic response appears to be very different from the responses we have observed, and less then 2% of the genes in which we detected alternative splicing overlap with those found in the hypoxic signature of HUVECs. It is possible that this is due to methods used (exon arrays versus our sequencing approach), cell type (HUVECs versus murine aortic endothelial cells), or

pathway-specific responses. The latter is likely, since we observed no consistent regulation of hypoxia-induced genes (e.g. *Vegfa*) in the flow-responsive signature. Alternative splicing responses are distinct, similar to transcriptional responses (*Graveley et al., 2011*). Indeed, even within the alternative-splicing response to altered flow, we are able to detect a distinct response dependent on the recruitment of circulating hematopoietic cells.

## Biological consequences of alternative splicing programs induced by low flow

We do not yet know the function of most of the splicing changes we have observed. Unlike mice deficient in both *EIIIA* and *EIIIB*, which exhibit an increased risk of arterial rupture under low flow, *Rbfox2 EC-KO* mice do not. This may be due to the lesser effect of *Rbfox2* deletion on *EIIIA* than *EIIIB* in the *Rbfox2* mutant mice (*EIIIA* inclusion was 50% in *Rbfox2 EC-KO* low-flow artery, versus 60% in WT low-flow artery, compared with 20% in high flow artery; while *EIIIB* inclusion was absent in all *Rbfox2 EC-KO* conditions). In previous work, we found that *EIIIA-/-* replicated much of the effect of the *EIIIAB-/-* mice; loss of *EIIIB* may have other consequences than loss of *EIIIA* (*Murphy and Hynes, 2014*).

Nevertheless, we have reason to believe that these splicing changes are biologically important. First, the alternatively spliced exons comprising this response, including *Fn-EIIIA* and *-EIIIB*, are highly conserved among all placental mammals. In fact ~80% of the flow-regulated skipped exons are as well conserved as consensus coding exons (CCDS). Conservation of sequence is not necessarily the same as conservation of regulation (*Kalsotra et al., 2008*; *Li et al., 2015*; *Merkin et al., 2012*; *Pai et al., 2016*; *Yeo et al., 2005*). Nevertheless, the conservation of regulatory regions, specifically the Rbfox2 motif, surrounding the skipped exons support conserved regulation as well. Second, deletion of a regulator of this splicing program, *Rbfox2*, affects the transcriptional response of the intima to low flow. Notably, both genes with *Rbfox2*-regulated transcript levels, and those with *Rbfox2*-regulated skipped exons were enriched in functions related to cytoskeletal remodeling, cell adhesion, and antigen presentation (*Figure 1* and *Figure 5*). The particular genes regulated in each way (transcript abundance and splicing) did not directly overlap, suggesting that similar biological processes may be targeted through both transcriptional and post-transcriptional mechanisms in *Rbfox2 EC-KO* mice. Given the known activation of TLR4 signaling by EIIIA (*Okamura et al., 2001*), a key player in adaptive immunity, the potential impact on antigen presentation and the adaptive response warrants further research.

## Regulation of alternative splicing in the aortic endothelium by platelets and macrophages

Much of the work to date on flow-induced changes in gene expression in the arterial endothelium has focused on the initial response to flow, which increases the transcriptional expression of adhesion receptors and chemokines involved in the recruitment of circulating immune cells (*Conway and Schwartz, 2013*; *Ley et al., 2007*). This has been a consequence of efforts to identify the first events leading to endothelial dysfunction under low flow. However, once circulating hematopoietic cells are recruited, they initiate ongoing interactions in the arterial wall likely to affect the ultimate outcome of flow-induced arterial injury (*Pober and Sessa, 2007*). Flow-driven vascular inflammation occurs as a chronic process, involving components of the innate and adaptive immune systems (*Galkina and Ley, 2009*; *Pober and Sessa, 2007*). Here, we show that recruitment of platelets and monocytes/macrophages in particular potently regulate alternative splicing in the arterial endothelium. Within this response, we observed alterations in a number of pathways that may impact ongoing interactions at the arterial wall. Thus, subsequent to the well-studied events leading to recruitment of immune cells, their interactions with the arterial endothelium appear to be affecting pathways that may help to shape the chronic inflammatory response.

## Endothelial functions of Rbfox2

Rbfox2 is one of three Rbfox family members (Rbfox-1,-2 and -3). The Rbfox family of splice factors and the motif recognized by their RNA-binding domain are conserved throughout the vertebrates examined and also in *C. elegans* (*Underwood et al., 2005*). The motif which the Rbfox family recognizes was first identified adjacent to *Fn-EIIIB* (*Huh and Hynes, 1993*, *1994*), which we find to be

dependent on Rbfox2 in our system, consistent with previous work in other systems (*Jangi et al., 2014*). The Rbfox family has been shown to have important functions in heart (*Gao et al., 2016*), muscle (*Singh et al., 2014*), and brain development (*Gehman et al., 2012*) and EMT in cancer (*Shapiro et al., 2011*). De novo mutations in *Rbfox2* have also been identified in congenital heart disease (*Homsy et al., 2015*), and alterations in Rbfox2 activity may be an early event in diabetes-induced heart pathology (*Nutter et al., 2016*). In both heart and brain, Rbfox2 has been shown to have a requirement in maintenance of tissue functions (*Gao et al., 2016*; *Gehman et al., 2012*). In contrast to the other tissues mentioned, *Rbfox2* is the only member of the Rbfox family expressed in the arterial endothelium.

How *Rbfox2* modulates splicing activity in this system remains unknown. We observe modest changes in expression of Rbfox2 (<20% increase in RNA in arterial endothelium exposed to low flow in vivo). Similar modest changes were observed in the in vitro system, in which we induced splicing changes with co-culture of platelets, monocytes and plasma. How else might Rbfox2 splicing activity be regulated? One possibility we examined is that upstream cues from platelet and monocyte recruitment alter sub-cellular localization of the Rbfox2 protein. However, we observed no obvious loss in nuclear Rbfox2 localization in vitro under conditions in which splicing changes are induced (*Figure 4—figure supplement 2*), suggesting that activity may be regulated in some other way. These data, together with the observation that deletion of *Rbfox2* results in splicing alterations more widespread than the low-flow response alone (*Figure 4D*), suggest that the specificity of the endothelial splicing response is driven by other flow-responsive factors acting in an *Rbfox2*-dependent manner.

We have focused on the splicing functions of *Rbfox2*. However, in addition to affecting splicing patterns, *Rbfox2* has also been reported to regulate gene expression patterns by other mechanisms. Splicing of Rbfox2 targets has been shown to regulate transcript stability (*Jangi et al., 2014*). Through recruitment of polycomb complexes to DNA, Rbfox2 may affect transcript expression (*Wei et al., 2016*). Rbfox2 may also stabilize transcripts directly by binding to the 3'UTR (*Damianov et al., 2016*; *Lee et al., 2016*), suppressing the formation of isoforms destined for nonsense-mediated decay (*Jangi et al., 2014*). However, as we observed no enrichment of known Rbfox-binding sites in the 3'UTR of regulated transcripts and no regulation of transcript levels in vitro which were regulated in vivo, we believe that many of the transcript level changes mediated by deletion of Rbfox2 are due to downstream or non-cell autonomous effects. Future work, examining the functional effects of individual splicing events identified will be important in teasing apart splicing-dependent and -independent effects.

In conclusion, we report that hundreds of splicing changes are regulated in endothelial cells exposed to an acute reduction in blood flow, many of which were dependent on recruitment of platelets and monocytes to the arterial wall and impact key pathways in endothelial cell biology. Rbfox2 emerged as an important regulator in this system, and its depletion resulted in a partial suppression of transcriptional responses characteristic of low-flow-induced endothelial activation. Thus, *Fn* is only one of many genes affected by splicing changes in the activated vascular endothelium. Alterations in *Fn* splicing contribute to a variety of vascular diseases with an inflammatory component; stroke (*Dhanesha et al., 2015*), atherosclerosis (*Tan et al., 2004*), myocardial infarction (*Arslan et al., 2011*), organ transplant and other fibroses (*Bhattacharyya et al., 2014*; *Booth et al., 2012*). Therefore, further investigation of the other alternative splicing events we have identified here may uncover novel regulatory mechanisms in chronic inflammatory disease.

## Materials and methods

**Key resources table**

| Reagent type (species) or resource | Designation | Source or reference | Identifiers |
| --- | --- | --- | --- |
| antibody | anti-GP1bα | Emfret | RRID:AB_2721041 |
| antibody | anti-Gr1 | Biolegend | RRID:AB_467731 |
| antibody | anti-Rbfox2 | Bethyl Labs | RRID:AB_609476 |
| commercial assay or kit | SMARTer Universal Low Input RNA Kit | Clontech | |

*Continued on next page*

*Continued*

| Reagent type (species) or resource | Designation | Source or reference | Identifiers |
|---|---|---|---|
| commercial assay or kit | RNAeasy microcolumns | Qiagen | |
| software, algorithm | RSEM v.1.2.15 | GitHub | RRID:SCR_013027 |
| software, algorithm | MISO v.0.4.9 | GitHub | RRID:SCR_003124 |
| software, algorithm | STAR 2.5.1b | GitHub | RRID:SCR_015899 |
| software, algorithm | Tophat 2.0.6 | GitHub | RRID:SCR_013035 |
| software, algorithm | Bowtie2 v. 2.0.5 | GitHub | RRID:SCR_005476 |
| software, algorithm | GSEA | http://www.gsea-msigdb.org/gsea/index.jsp | RRID:SCR_003199 |
| strain, strain background (*M. musculus*) | *Rbfox2^{lox/lox}* | Jackson Lab | RRID:IMSR_JAX:014090 |
| strain, strain background (*M. musculus*) | *Rosa26-mTmG* | Jackson Lab | RRID:IMSR_JAX:007676 |
| strain, strain background (*M. musculus*) | *Cdh5(PAC)-CreERT2* | Jackson Lab | RRID:IMSR_TAC:13073 |
| strain, strain background (*M. musculus*) | C57BL/6J | Jackson Lab | RRID:IMSR_JAX:000664 |

## Flow alterations

Partial carotid ligations were performed as previously described (*Ni et al., 2010*), with minor modifications (*Murphy and Hynes, 2014*). Briefly, the distal branches of the left carotid artery were identified in mice anesthetized with isoflurane. The left external carotid, internal carotid, and occipital artery were ligated with 9–0 Ethilon suture, leaving only the superior thyroid artery intact. Sham operations consisted of the same carotid dissection and encircling with suture, except that the vessels were not tied off. High-resolution ultrasound, using the VisualSonics Vevo 770, was performed at the experimental endpoint (2–7 days after partial carotid ligation) to confirm vessel patency.

## RNA-isolation and sequencing

Isolation of intimal RNA was performed as previously described from the carotid intima (*Murphy and Hynes, 2014*; *Ni et al., 2010*). Briefly, vessels were isolated and imaged from euthanized mice, flushed with PBS solution and then 150 µL Trizol. RNA from cultured aortic endothelial cells was isolated after one week in culture by sorting 1–2K endothelial cells (by CD31+ and Icam2+) from BD Aria directly into cold Trizol solution. RNA from all samples was isolated with Qiagen RNAeasy microcolumns. Pools of RNA were created and then DNAse-treated in solution, before concentrating on RNAeasy columns with elution in minimal volume. RNA quality was assessed by Agilent Bioanalyzer or Advanced Analytical, library preparation was performed using the Clontech SMARTer Universal Low Input RNA Kit, and samples were sequenced on Illumina HiSeq 2000 or NextSeq 500 using 80 bp paired-end (PE) reads (triplicates of low-flow, sham and high-flow), 100 bp PE reads (re-sequencing of triplicates of low-flow, sham and high-flow, and cultured aortic endothelial cells) or 150 bp PE reads (hematopoeitic cell depletions with controls and endothelial Rbfox2 deletion with controls in vivo and in vitro). Base calls were performed using the Offline Base Caller (Illumina) v. 1.9.4 and reads mapping to different genomic features were tallied and read densities were compared for each sample.

## Read-mapping and transcript analysis

For experiments with alteration of flow (triplicates of low-flow, sham and high-flow), reads were trimmed to 80 bp and mapped with Tophat 2.0.6 and Bowtie2 v. 2.0.5, allowing the detection of novel junctions, with options -p 8 –read-edit-dist 4 –min-intron-length 10 –read-mismatches 4 –max-intron-length 1000000 –read-realign-edit-dist 0 –mate-std-dev 30 –segment-length 20 –library-type fr-firststrand -r 50, guided (-G) by an mm9-based refseq gtf file [https://www.ncbi.nlm.nih.gov/pubmed/23618408].

For experiments with alteration of flow with or without depletion of hematopoeitic cells or deletion of Rbfox2, 150 bp reads were mapped with STAR 2.5.1b and options –runThreadN 8 –runMode alignReads –outFilterType BySJout –outFilterMultimapNmax 20 –alignSJoverhangMin 8 –alignSJD-BoverhangMin 1 –outFilterMismatchNmax 999 –alignIntronMin 10 –alignIntronMax 1000000 –

alignMatesGapMax 1000000 –outSAMtype BAM SortedByCoordinate –quantMode Transcriptome-SAM [https://www.ncbi.nlm.nih.gov/pubmed/23104886].

RSEM v.1.2.15 (options –paired-end –calc-ci -bam -p 8 ) was used to calculate transcript levels [https://www.ncbi.nlm.nih.gov/pubmed/21816040], and DESeq2 v1.10.1 in the R statistical environment v3.2.3 was used to calculate differential transcript levels using the DEseq() integrated command with default parameters, which estimates library sizes and dispersions and fits a negative binomial generalized linear model for fold-changes and Wald statistics estimation using a 2-condition design. [https://www.ncbi.nlm.nih.gov/pubmed/25516281].

For *eGFP* and *Tomato* analysis, a custom index containing *eGFP* and *tdTomato* sequences was created, and bowtie was used to align reads to this index using options –best –strata -m 1 –q. The number of reads mapping to *eGFP* and to *tdTomato* were counted, and % tdTomato reads determined [*tdTomato* reads / (*tdTomato*+ *eGFP* reads) *100%].

## Splicing analysis

Positive control data sets for regulation of splicing by *Rbfox2 in vitro* or by *Mbnl in vivo* were taken from published data, and run through our own informatics pipeline (*Jangi et al., 2014*; *Wang et al., 2015*).

MISO v.0.4.9 was used to determine differential splicing regulation. The SE, A3SS, A5SS, RI and MXE event index was derived from the MISO splicing events database (mm9 version 2). Briefly, this set of events was derived by considering all transcripts annotated in Ensembl genes, knownGenes (UCSC) and RefSeq genes (as of June 2013). The ALE and TandemUTR events index was derived from the MISO database compiled from published data, made available on the MISO website (*Hoque et al., 2013*). Additional novel SE, A3SS, A5SS and MXE events were detected using a method developed by Paul Boutz. Identification and classification of novel alternative splicing events was performed as described in *Boutz et al. (2015)*. Briefly, mapped splice junctions were filtered for minimal transcript level and classified as SE, A3SS, A5SS, or MXE without reference to annotated exonic loci. This set of novel events was then used to generate a novel index, which was passed through the same MISO pipeline as all other indices.

## Selection of subgroups of regulated exons

*Flow-regulated*: BF >5 in two completely independent biological comparisons of low flow versus high flow, where the difference between high flow and low flow is >2 x the difference between replicate high-flow conditions.

### Flow-regulated and endothelial

Of the events above, at least 50% of the difference between low-flow in vivo and high flow in vivo must be recapitulated in differences between FACs-purified mouse arterial cells in culture and high-flow cells in vivo.

### Flow-regulated and endothelial and platelet-regulated

Of the events above, if platelet depletion reverted flow-induced splicing changes by >50% in two completely independent biological comparisons.

### Flow-regulated and endothelial and platelet-regulated and in vitro regulated

Of the events above, if the change in splicing induced in isolated aortic endothelial cells by the addition of platelets, macrophages and plasma was >50% of the flow-regulated change in splicing.

### Rbfox2-regulated

If change in inclusion with Rbfox2 deletion was BF >5 in any of three comparisons, in vivo low-flow Rbfox2 control vs. Rbfox2 EC-KO artery at 7 days, in vivo high-flow Rbfox2 control vs. Rbfox2 EC-KO artery at 7 days, or in vitro Rbfox2 control vs. Rbfox2-deleted aortic endothelial cells, and consistently changed in the same direction in all in vivo comparisons under low flow between Rbfox2 control and Rbfox2 EC-KO arteries. In this set, Rbfox2 control arteries have floxed alleles but no Cre activity.

## Identification of enriched splice factor motifs adjacent to regulated skipped exons

To perform motif enrichment analysis, we compared endothelial alternative skipped exons regulated by blood cells in vivo and in vitro (the foreground) with skipped exons expressed by endothelial cells, but not consistently regulated under the same conditions (the background). In general, the foreground events were more often poised at an intermediate Psi level (neither complete inclusion or exclusion). They also tended to be expressed at a higher level than background events. Thus, we created background sets in which we normalized each of these parameters independently (data not shown). We looked for motifs enriched either upstream (3'splice site, or 3'SS) or downstream (5' splice site, or 5'SS) of the regulated exons. Since the direction of change in exon inclusion, increased dPsi or decreased dPsi, has been correlated with the location of splice factor binding, we further separated the foreground into exons with increased (+dPsi) or decreased (-dPsi) inclusion. Intronic sequences adjacent to the 3'SS and the 5'SS of skipped exons with increased or decreased inclusion were isolated from mm9 sequence data using Bedtools 2.16.1 (getfasta –s), and enrichment was calculated based on enrichment of 6-mer or 7-mer motifs in the foreground versus the background sets, split across 10 GC-normalized bins, iterated multiple times.

From the lists of enriched motifs we obtained, we aimed to identify the likely RNA-binding proteins. To do this in an unbiased manner, we used the CisBP-RNA data set, which includes 154 unique motifs and 373 total murine RNA-binding proteins (*Ray et al., 2013*). We modified the ranking algorithm GSEA, typically used to identify transcriptional signatures, to identify RNA-binding protein signatures. A Kolmogorov-Smirnov statistic embedded in GSEA was used to identify which of these RNA-binding-protein motif sets was most abundant at the leading edge of the ranked enrichment set (*Subramanian et al., 2005*).

## Depletion of innate immune cells

For platelet depletion experiments, mice were injected intraperitoneally with a single dose of platelet-depleting antibody (anti-GP1bα; 50 μg per mouse; R300; Emfret; RRID:AB_2721041) or IgG control at the time of partial carotid ligation. To deplete Ly6G+ cells, anti-Gr1 (RB6-8C5; BioLegend; RRID:AB_467731) or IgG controls were administered by intraperitoneal injection (25 μg per mouse) 24 hr prior to partial carotid ligation. To deplete macrophages, mice were injected by tail vein with 150 μL clodronate liposomes or PBS liposome controls at the time of partial carotid ligation. Clodronate liposomes or PBS liposomes were purchased from ClodronateLiposomes.com (Kruisweg 59, 2011 LB Haarlem, The Netherlands).

## Mice

Male C57BL/6J mice from Jackson Laboratories were used at 6–7 weeks of age for experiments on the regulation of transcription and splicing in the carotid intima in wild-type mice, or in wild-type mice with the depletion of various hematopoietic cells.

For the endothelial deletion of Rbfox2, *Cdh5(PAC)-CreERT2, Rosa26-mTmG* and *Rbfox2^{lox/lox}* mice have been previously described (*Gehman et al., 2012*; *Muzumdar et al., 2007*; *Wang et al., 2010*) RRID:IMSR_JAX:014090, RRID:IMSR_JAX:007676, RRID:IMSR_TAC:13073. They were intercrossed to create the Rbfox2 EC-KO mice *(Cdh5(PAC)-CreERT2; Rosa26-mTmG; Rbfox2^{lox/lox})* and littermate controls (*Rosa26-mTmG* and *Rbfox2^{lox/lox}*) used here. Mice used were between 2 and 7 months of age in paired groups of males and females.

All mice were housed and handled in accordance with protocols approved by the Massachusetts Institute of Technology Division of Comparative Medicine.

## Aortic endothelial cell isolation

Mouse aortic endothelial cells were isolated following previously described methods (*Kobayashi et al., 2005*), with modifications. Briefly, mice were perfused via the left ventricle with PBS, the aorta was isolated and filled with 2% collagenase II (Worthington) in serum-free DMEM, and then closed at the ends with 7–0 suture. The vessel was digested for 30–45 min in 10% FBS at 37C, and endothelial cells were flushed out into EC culture medium, and onto a collagen I-coated plate. EC culture medium consisted of DMEM with 10% FBS and 10 mg/mL endothelial growth supplement (ECGS, Biomedical Technologies) with primocin (InvivoGen). Typical isolations of ~1000 cells

expanded to ~10,000–50,000 cells. Cells were then FACs-sorted (BD Aria) on endothelial markers, eGFP for *Cdh5(PAC)-CreERT2; Rosa26-mTmG* on Pecam+ and Icam2+ for *Cdh5(PAC)-CreERT2; Rosa26-mTmG; Rbfox2$^{lox/lox}$* mice and littermate controls.

## Aortic endothelial cell immortalization

After purification by FACs, endothelial cells were immortalized by lentiviral TetOn-Sv40T. The construct was developed from a version of pTRIPZ (Clontech) with removal of the puro-selection cassette. SV40 T antigen was inserted into the tet-regulated region of the lentivirus from pBabe-SV40T (*Zhao et al., 2003*). After infection, cells were expanded in EC media with 2 μg/mL Doxycycline (Dox).

## Aortic endothelial cell co-culture

Cells derived from *Cdh5(PAC)-CreERT2; Rosa26-mTmG*, tamoxifen-treated in vivo, and then isolated by FACs purification on eGFP, expanded and immortalized by TetOn-SV40T were trypsinized and split in DMEM into new dishes, without Dox to turn off SV40T. After culture for 24 hr in DMEM, combinations of platelets, bone-marrow-derived monocytes, and/or 10% plasma were added for 48 hr. Cells in plate were then imaged and lysed with Trizol for RNA extraction.

Platelets were isolated from the platelet-rich plasma (PRP) of C57BL/6J mouse blood by collection in ACD buffer. They were spun down at 500xG for 7 min with isolation of the upper PRP, from which a platelet pellet was isolated after being spun down at 2800xG for 5 min and resuspended in PIPES buffer. ~20 million platelets (in 6 μL PIPES) were added to each confluent well of endothelial cells in the 24-well plate. Plasma was taken from C57BL/6J mouse blood collected in EDTA and spun down at 2000xG for 15 min at 4°C. 10% plasma was added to DMEM in 24-well plates. Bone-marrow-derived monocytes were isolated from the femurs of C57BL/6J mice, flushed with 10% FBS DMEM (+2 mM EDTA) into collection tube with 10 mL syringe and 25Ga needle, and then purified using an EasySep Mouse Monocyte Isolation Kit (by negative selection, depleting Cd3+, Cd45R+, Cd117+, Ly6G+, Nk1.1+, Siglec F+ cells). ~10K monocytes were added to each well of the 24-well plate.

## Aortic endothelial cell immunofluorescence

Cells (TetOn-Sv40, eGFP+ from *Cdh5(PAC)CreERT2; mT/mG* mice were cultured on poly-L-lysine treated coverslips and treated or not with 10% serum (FBS) for 48 hr. Coverslips and cells were washed with phosphate buffered saline (PBS) and fixed with ice cold methanol. Rbfox2 protein was stained by anti-Rbfox2 (Bethyl Labs A300-864A, 1:1000; RRID:AB_609476), with goat anti-rabbit Alexa594 secondary. Block was 10% normal goat serum in PBS with 0.1% tritonX-100.

## Acknowledgements

We thank members of the Hynes lab and other labs in the Koch Institute for advice and discussions, particularly Mohini Jiangi for discussions of Rbfox2 functions and John Lamar (Albany Medical Center) for critical reading. We thank the Swanson Biotechnology Center at the Koch Institute/MIT, especially the Applied Therapeutics and Whole Animal Imaging Facility, the Hope Babette Tang (1983) Histology Facility, the Microscopy Facility, the Barbara K Ostrom (1978) Bioinformatics and Computing Facility, and Scott Malstrom, Denise Crowley, Eliza Vasile and Charlie Whittaker for technical support. The authors wish to dedicate this paper to the memory of Officer Sean Collier for his caring service to the MIT community.

## Additional information

### Funding

| Funder | Grant reference number | Author |
| --- | --- | --- |
| National Heart, Lung, and Blood Institute | F32-HL110484 | Patrick A Murphy |
| National Cancer Institute | P30-CA14051 | Vincent L Butty |

| Howard Hughes Medical Institute | Investigator Award | Richard O Hynes |
| National Heart, Lung, and Blood Institute | K99/R00-HL125727 | Patrick A Murphy |
| National Heart, Lung, and Blood Institute | PO1-HL66105 | Patrick A Murphy |
| National Institute of General Medical Sciences | R01-GM034277 | Phillip A Sharp |

The funders had no role in study design, data collection and interpretation, or the decision to submit the work for publication.

## Author contributions

Patrick A Murphy, Conceptualization, Resources, Data curation, Software, Formal analysis, Funding acquisition, Validation, Investigation, Visualization, Methodology, Writing—original draft, Project administration, Writing—review and editing; Vincent L Butty, Data curation, Software, Formal analysis, Supervision, Methodology, Writing—review and editing; Paul L Boutz, Data curation, Software, Formal analysis, Investigation, Methodology, Writing—review and editing; Shahinoor Begum, Amy L Kimble, Investigation; Phillip A Sharp, Resources, Software, Funding acquisition, Writing—review and editing; Christopher B Burge, Conceptualization, Methodology; Richard O Hynes, Conceptualization, Resources, Supervision, Funding acquisition, Methodology, Project administration, Writing—review and editing

## Author ORCIDs

Patrick A Murphy (iD) https://orcid.org/0000-0002-2956-1042
Christopher B Burge (iD) http://orcid.org/0000-0001-9047-5648
Richard O Hynes (iD) http://orcid.org/0000-0001-7603-8396

## Ethics

Animal experimentation: All mice were housed and handled in accordance with protocols approved by the Massachusetts Institute of Technology Committee on Animal Care (CAC) protocol (0415-033-18). All surgery was performed under isoflurane anesthesia with post-operative analgesia.

## Decision letter and Author response

Decision letter https://doi.org/10.7554/eLife.29494.043
Author response https://doi.org/10.7554/eLife.29494.044

# Additional files

## Supplementary files

• Supplementary file 1. Usage of Source data.
DOI: https://doi.org/10.7554/eLife.29494.032

• Transparent reporting form
DOI: https://doi.org/10.7554/eLife.29494.033

## Major datasets

The following dataset was generated:

| Author(s) | Year | Dataset title | Dataset URL | Database, license, and accessibility information |
|---|---|---|---|---|
| Murphy PA, Butty VL, Boutz PL, Begum S, Sharp PA, Burge CB, Hynes RO | 2017 | Alternative RNA Splicing in the Endothelium Mediated in Part by Rbfox2 Regulates the Arterial Response to Low Flow | https://www.ncbi.nlm.nih.gov/geo/query/acc.cgi?acc=GSE101826 | Publicly available at the NCBI Gene Expression Omnibus (accession no: GSE101826) |

The following previously published datasets were used:

| Author(s) | Year | Dataset title | Dataset URL | Database, license, and accessibility information |
|---|---|---|---|---|
| Wang ET, Cody NA, Jog S, Biancolella M, Wang TT, Treacy DJ, Luo S, Schroth GP, Housman DE, Reddy S, Lécuyer E, Burge CB | 2012 | Transcriptome-wide Regulation of Splicing and mRNA Localization by Muscleblind Proteins | https://www.ncbi.nlm.nih.gov/geo/query/acc.cgi?acc=GSE39911 | Publicly available at the NCBI Gene Expression Omnibus (accession no: GSE39911) |
| Jangi M, Boutz PL, Paul P, Sharp PA | 2014 | Rbfox2 controls autoregulation in RNA binding protein networks | https://www.ncbi.nlm.nih.gov/geo/query/acc.cgi?acc=GSE54794 | Publicly available at the NCBI Gene Expression Omnibus (accession no: GSE54794) |

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
