## [Decision Letter]

Thank you for submitting your article "Alternative RNA Splicing in the Endothelium Mediated in Part by Rbfox2 Regulates the Arterial Response to Low Flow" for consideration by *eLife*. Your article has been favorably evaluated by James Manley (Senior Editor) and two reviewers, one of whom, Douglas Black, is a member of our Board of Reviewing Editors..

The reviewers have discussed the reviews with one another and the Reviewing Editor has drafted this decision to help you prepare a revised submission.

Summary:

This paper from the Hynes group examines the role of alternative splicing in the response of arterial endothelium to low blood flow. Endothelial recruitment of inflammatory cells in response to low flow is important in a variety of vascular diseases including atherosclerosis. Earlier work identified endothelial gene expression changes induced by the recruitment of platelets and additional cell types to sites of low flow and also showed that changes in the splicing of fibronectin (Fn) were important. Here the authors do a broader analysis, using carotid ligation to model low flow in the mouse and applying RNAseq to identify changes in splicing pattern across the transcriptome. This identified a large number of altered transcripts. They develop similar results in cultured endothelial cells and use antibody depletion of platelets or macrophages to identify the set of splicing changes responsive to the recruitment of these cells, which include the aforementioned Fn. Examining the cassette exons among the platelet/macrophage responsive splicing events, they find enrichment for the binding site of Rbfox family splicing regulators in the introns adjacent to the flow regulated exons. They show that Rbfox2 is expressed in the endothelium and use a conditional Rbfox2 allele to conditionally knock it out of these cells with the Cdh5-Cre driver. Examining the splicing in the knockout mice, they find that a substantial subset of the low flow responsive exons require Rbfox2 in their regulation. Finally, they find that many of the previously characterized gene expression changes in low flow conditions also require Rbfox2. These data point to a new posttranscriptional genetic regulatory program mediating endothelial responses to blood flow.

The reviewers all found this paper to be novel and interesting. Alternative splicing and the program of Rbfox regulation specifically are likely to be important for both normal and disease-altered mechanisms of vascular function. While the conclusions in the paper are supported by the data, the data are exclusively transcriptome analysis, limiting the potential mechanistic and biological insights into questions such as what signals from macrophages or platelets stimulate the transcriptome-wide effects; how Rbfox2 coordinates, through changes in activity or concentration, the low flow-induced changes in endothelial cells; or what physiologic impact the Rbfox2 knockout has on the response to low flow conditions. Addressing such questions would greatly broaden the interest of the paper.

Essential revisions:

1) Do the Rbfox2 ECKOs have any noticeable phenotype? For instance, do these mice exhibit an altered response to low flow when compared to littermate controls? Transcriptome-wide changes are presented but are not investigated further. The only physiologic data presented for Rbfox2 ECKOs is expression of macrophage markers. If the mutant phenotype has already been examined in more detail without finding a major difference from wildtype, this should be discussed.

2) As is alluded by the authors in the Discussion, the immune response to low flow occurs rapidly. However, the disease progression involves a chronic inflammatory response. The authors should check if the initial transcriptome changes persist in the wildtype, immune response depleted, and Rbfox2 EC-KOs over a longer time period. Another round of RNA-seq is not necessary, but additional verification by qPCR and splice assays of key events will be a valuable addition.

3) It is unclear how Rbfox2 activity responds to low flow since the authors state that Rbfox2 mRNA expression changes modestly under these conditions. Have they confirmed this at the protein level? Moreover, Rbfox proteins have both nuclear and cytoplasmic isoforms, with the cytoplasmic forms regulating transcript levels through 3' UTR binding. The authors should assess how much Rbfox2 is in the cytoplasm of endothelial cells and whether this cytoplasmic Rbfox2 is causing some of the changes in gene expression observed in low flow conditions. This would yield a more complete picture of the post-transcriptional endothelial regulatory program induced by low flow.

---

## [Author Response]

Essential revisions:1) Do the Rbfox2 ECKOs have any noticeable phenotype? For instance, do these mice exhibit an altered response to low flow when compared to littermate controls? Transcriptome-wide changes are presented but are not investigated further. The only physiologic data presented for Rbfox2 ECKOs is expression of macrophage markers. If the mutant phenotype has already been examined in more detail without finding a major difference from wildtype, this should be discussed.

We examined a number of responses in the Rbfox2-ECKO mice in the acute low flow model, including the transcriptional responses described in the paper. We have not observed obvious phenotypes in arterial hemorrhage, neointima growth, or differences in the recruitment of Cd68+ or F4/80+ macrophages or Cxcr2+ neutrophils, based on cell-specific RNA transcripts in the intima in this model. These findings are now described in the Discussion (subsection “Biological consequences of alternative splicing programs induced by low flow”). We predict, based on the intimal transcriptional changes observed (Figure 5), that adaptive responses could be affected. We have additional data to support this hypothesis; see Author response image 1, which reveals that culture on Rbfox2-ECKO endothelial cells shows differential activation of a number of genes in monocytes as compared with culture on control endothelial cells (e.g. Tmem176a, Tmem176b, Gpx3, Hpgd). Tmem176b is also known as Torid (tolerance-related and induced), and along with Tmem176a is downregulated during dendritic cell differentiation in response to LPS, poly I:C, and overexpression of Tmem176b suppresses MHCII expression on LPS-treated bone marrow derived monocytes (PMID: 20501748, 16095493). We have not included these preliminary data in the revised manuscript since they require further extension, which requires modifications to the disturbed flow model to promote the adaptive response. This work is underway, but the establishment of these models and their characterization will take more time and is beyond the scope of this paper.

**Author response image 1. respfig1:** Endothelial cells deficient in Rbfox2 alter the response of co-cultured monocytes and replicate aspects of the in vivo intimal response. Wild-type bone marrow monocytes were isolated and cultured for 5 days on Rbfox2-ECKO or control aortic endothelial cells (N=5 and N=7). Transcriptome of the initial monocytes and the co-cultured monocytes was analyzed by RNA-seq. (**A**) Overlap of the genes with pvalue<0.05 by DESeq2 with the genes identified by the same threshold in vivo. (**B**) Comparison of the fold-change observed in vitro, in co-cultured monocytes, and in vivo, in the intimal flush of low-flow carotid arteries. In red/pink are genes concordant in both, in blue are genes with opposing changes in vivo and in vitro. (**C**) Log2 fold-change (relative to the input monocytes at Day0) in monocytes cultured on biological replicates of Rbfox2-ECO or WT aortic endothelial cells.

2) As is alluded by the authors in the Discussion, the immune response to low flow occurs rapidly. However, the disease progression involves a chronic inflammatory response. The authors should check if the initial transcriptome changes persist in the wildtype, immune response depleted, and Rbfox2 EC-KOs over a longer time period. Another round of RNA-seq is not necessary, but additional verification by qPCR and splice assays of key events will be a valuable addition.

We agree with the reviewers that an examination of the effects of innate immune cell recruitment and Rbfox2 regulation on the chronic low-flow injury response is warranted. Following their suggestion, and our own interest, we have depleted platelets from the low flow artery two weeks after the induction of low flow, and found that this late depletion was insufficient to revert EIIIA splicing changes, or PAI1 expression, a marker of endothelial cell activation (Author response image 2). However, this late-stage response is complex. In the absence of platelets, both PAI1 and EIIIA expression appeared elevated, suggesting a role at this stage which opposes the earlier acute functions of platelets. However, there are other components of this late response which also do not conform with the early response. Cd68, for example, is no longer elevated in the low flow artery. At this point, our understanding of immune-endothelial interactions during chronic disease progress is inadequate. More work will be required to determine why platelet depletion would exert different responses depending on the timing. Better, and more cost-effective models must be adapted to allow the chronic depletion of platelets in this response for several weeks, without the potential immune complications of long term antibody mediated depletion. As discussed above, ongoing work is focused on the establishment of these models, and an examination of the role of splicing changes in the chronic inflammatory response. The current paper does not directly address this, and will not be able to do so without new models and substantial additional work, so again we have not included these preliminary data in the revised manuscript.

**Author response image 2. respfig2:** Late platelet depletion in chronic low flow injury is insufficient to reverse EIIIA splicing and PAI1 expression. Wild-type mice were subjected to partial carotid ligation, and two weeks later platelets were depleted or not. Six days after depletion, intimal RNA flushes were obtained, and levels of gene expression assessed by qPCR.

3) It is unclear how Rbfox2 activity responds to low flow since the authors state that Rbfox2 mRNA expression changes modestly under these conditions. Have they confirmed this at the protein level? Moreover, Rbfox proteins have both nuclear and cytoplasmic isoforms, with the cytoplasmic forms regulating transcript levels through 3' UTR binding. The authors should assess how much Rbfox2 is in the cytoplasm of endothelial cells and whether this cytoplasmic Rbfox2 is causing some of the changes in gene expression observed in low flow conditions. This would yield a more complete picture of the post-transcriptional endothelial regulatory program induced by low flow.

Given the challenges of staining of nuclear vs. cytoplasmic protein in the very thin endothelial cell layer in vivo, or specifically isolating protein from this layer, we have examined cellular localization of Rbfox2 in vitro, where splicing is regulated as in vivo by the addition of monocytes, platelets and plasma or by the serum derived from their combination. In endothelial cells in vitro, Rbfox2 protein appears primarily nuclear by immunofluorescence, with no discernable change upon treatment with serum. These data are now presented as Figure 4—figure supplement 2, and described in the Results (subsection “Endothelial-specific deletion of Rbfox2 partially reverts flow-responsive skipping of vascular skipped exons induced by low flow in vivo”, fourth paragraph) and Discussion (subsection “Endothelial functions of Rbfox2”, second paragraph). However, an important caveat to these in vitro experiments with regard to transcript regulation in the cytoplasmic pool is that, while in vivo Rbfox2-mediated splicing changes are replicated in vivo, changes in transcript levels were not. Thus, the in vitro system may not be positioned to define the contribution of cytoplasmic Rbfox2. Unfortunately, as mentioned earlier, the in vivo system is not amenable to this analysis either. Nevertheless, arguing against a direct regulation of transcripts by cytoplasmic Rbfox2, we find no enrichment in Clip-Seq-defined Rbfox binding sites in the 3’UTR of transcripts whose levels (not splicing) were Rbfox2 regulated. These analyses are now described in the Results (subsection “Endothelial deletion of Rbfox2 alters the intimal response to low flow”, second paragraph) and Discussion (subsection “Endothelial functions of Rbfox2”, third paragraph).